# Diversion of Carbon Flux from Sugars to Lipids Improves the Growth of an Arabidopsis Starchless Mutant

**DOI:** 10.3390/plants8070229

**Published:** 2019-07-17

**Authors:** Jilian Fan, Chao Zhou, Linhui Yu, Ping Li, John Shanklin, Changcheng Xu

**Affiliations:** 1Biology Department, Brookhaven National Laboratory, Upton, New York, NY 11933, USA; 2College of Agriculture, Shanxi Agricultural University, Taigu 030801, Shanxi, China

**Keywords:** starch, sugars, lipids, triacylglycerol, phospholipid:diacylglycerol acyltransferase1

## Abstract

Inactivation of ADP-glucose pyrophosphorylase1 (ADG1) causes a starchless phenotype in Arabidopsis. Mutants defective in ADG1 show severe growth retardation in day/night conditions but exhibit similar growth to wild type under continuous light, implying that starch plays an important role in supporting respiration, metabolism and growth at night. In addition to carbohydrates, lipids and proteins can serve as alternative respiratory substrates for the energy production in mature plants. To test the role of lipids in plant growth, we generated transgenic plants overexpressing phospholipid:diacylglycerol acyltransferase1 (PDAT1) in *adg1*. We found that PDAT1 overexpression caused an increase in both fatty acid synthesis and turnover and increased the accumulation of triacylglycerol (TAG) at the expense of sugars, and enhanced the growth of *adg1*. We demonstrated that unlike sugars, which were metabolized within a few hours of darkness, TAG breakdown was slow, occurring throughout the entire dark period. The slow pace of TAG hydrolysis provided a sustained supply of fatty acids for energy production, thereby alleviating energy deficiency at night and thereby improving the growth of the starchless mutants. We conclude that lipids can contribute to plant growth by providing a constant supply of fatty acids as an alternative energy source in mature starchless mutant plants.

## 1. Introduction

Plants break down sugars through respiration to fuel metabolism, growth, storage and maintenance throughout the day/night cycle. During the day, leaves carry out photosynthesis, which converts carbon dioxide and water into organic compounds using sunlight as an energy source. The prime end-product of photosynthetic carbon assimilation, triose phosphate, is partitioned into several central metabolic pathways, including those leading to the generation of carbohydrates, proteins and lipids [1]. In many plants, including Arabidopsis, a large fraction of triose phosphate formed during photosynthesis is used to synthesize starch, a glucose polymer, in the form of semi-crystalline insoluble granules in the chloroplast. At night, when photosynthesis ceases, starch is hydrolyzed to provide a constant supply of sugars for respiration [2,3,4,5]. The physiological importance of diurnal starch metabolism is clearly illustrated in mutants that are defective in starch synthesis [6,7] or degradation [8,9]. For example, an Arabidopsis starchless mutant defective in *ADP-glucose pyrophosphorylase1* (*adg1*) shows growth retardation under long days, stunted growth under short days, but similar growth rates to wild type under continuous night [6]. The growth inhibition of starchless mutants in day/night conditions has been attributed to nighttime carbon starvation [4,5], increased root respiration [10] and sugar-induced feedback inhibition of photosynthesis [11]. 

Lipids serve as the building blocks of cellular membranes, and the storage lipid triacylglycerol (TAG) is amongst the most energy-rich compounds that occur in nature. In Arabidopsis and many other plants, two parallel pathways compartmentalized either in the plastid or the endoplasmic reticulum (ER) are responsible for synthesizing the vast majority of cellular lipids, principally glycerolipids [12]. Fatty acids, the predominant components of glycerolipids are synthesized in the plastid. They can be incorporated into thylakoid membranes, including monogalactosyldiacylglycerol (MGDG) and digalactosyldiacylglycerol (DGDG), in the plastid via the endogenous plastid pathway. Alternatively, fatty acids can be exported from the plastid and metabolized in the ER to generate both membrane phospholipids and TAG storage lipids via the ER pathway. A fraction of phospholipids assembled in the ER re-enter the plastid to support the plastid pathway of thylakoid lipid synthesis. Because the substrate specificity of acyltransferases in the ER and plastids differs, lipids assembled via either the ER or the plastid pathway can be distinguished by the presence of acyl groups with chain lengths of 16 carbons (C16) or 18 carbons (C18) at the *sn-2* position, respectively [13].

In vegetative tissues such as leaves, most of the fatty acid flux through the ER pathway is directed toward membrane lipid synthesis [14]. Consequently, TAG does not accumulate to significant levels in leaves [15], despite the presence of high TAG synthetic [16] and hydrolytic [17] activities. In plants, the last step of TAG synthesis is mediated by two distinct types of acyltransferases: acyl-CoA:diacylglycerol acyltransferase and phospholipid:diacylglycerol acyltransferase (PDAT) [18]. Overexpressing PDAT1 caused an increase in fatty acid synthesis, an increased diversion of fatty acids from membrane lipids to TAG and hence a significant increase in TAG levels in leaves [14]. 

The metabolic breakdown of TAG is catalyzed by lipases including SUGAR-DEPENDENT 1 (SDP1) [19], and the resultant free fatty acids are imported into the peroxisome by PEROXISOMAL ABC TRANSPORTER 1 (PXA1) where they are metabolized via β-oxidation to acetyl-CoAs, which represent key metabolites for energy production in mitochondria [20]. Disruption of SDP1 or PXA1 significantly compromises the growth and development of *adg1* [21], suggesting that in the absence of starch, lipids are used as an alternative respiratory substrate for energy production. Here, we show that overexpression of *PDAT1* enhances carbon partitioning into lipids and improves the growth of the *adg1* starchless mutant. Together, our studies reveal a previously unrecognized role of lipid metabolism in diurnal energy homeostasis and plant growth under normal growth conditions.

## 2. Results

### 2.1. Overexpressing PDAT1 Improves the Growth of adg1

To further investigate the physiological roles of lipids, we introduced the *PDAT1* gene driven by the constitutive CaMV 35S promotor [14] into *adg1* by agrobacterium-mediated transformation. We obtained more than 20 PDAT1-overexpressing transgenic lines and chose two independent lines for detailed phenotypic analysis. Under a 16-h/8-h day/night cycle, both lines grew faster than *adg1,* albeit slower than the wild type (Figure 1A). Quantitative analysis showed that total shoot fresh weight was 11.7% and 13.3% higher in transgenic lines 3 and 19, respectively relative to *adg1* at six weeks, and these increases were statistically significant (Figure 1B). Total leaf TAG content measured at the end of the light period was 14.9- and 18.5-fold higher in line 3 and line 19, respectively, than that of *adg1* (Figure 2). Leaf TAG levels measured at the onset of the light period decreased by 67% and 64% in lines 3 and 19, respectively, compared with those in the same lines measured at the end of the light period, suggesting that most of the TAG accumulated during the day was degraded during the subsequent night, likely as an alternative source of energy for metabolism and growth at night in the starchless mutant. 

### 2.2. Overexpressing PDAT1 Enhance the Synthesis and Turnover of Fatty Acids 

To understand how PDAT1 overexpression increases TAG accumulation in leaves, we first measured the rate of fatty acid synthesis. To do this we fed detached growing leaves with ^14^C-acetate, an immediate precursor for fatty acid synthesis in the plastid. Under our experimental conditions, ^14^C-acetate was incorporated into fatty acids in a linear fashion for at least 60 min [14]. As expected from our previous findings [21], the rate of radiolabel incorporation into total leaf fatty acids was higher in *adg1* compared with wild type (Figure 3A). Overexpression of PDAT1 in *adg1* resulted in additional significant increases in the rate of fatty acid synthesis relative to *adg1* (Figure 3A). To test whether fatty acid turnover is also affected in PDAT1-overexpressing lines, a pulse-chase labeling experiment was performed, in which detached leaves were first incubated with^14^C-acetate to label fatty acids for 1 h. They were then thoroughly washed to remove unincorporated radiotracer and incubated for additional 3 d. The rate of loss of radiolabeled fatty acids was significantly increased in *adg1* relative to wild type. The expression of PDAT1 in *adg1* resulted in additional significant increases in the rate of loss of label in lines 3 and 19 compared with *adg1* (Figure 3B), mirroring the observed increases in the rates of fatty acid synthesis seen in Figure 3A. Together, these radiotracer labeling results suggest that an overexpression of PDAT1 enhances both fatty acid synthesis and fatty acid turnover to similar extents in the *adg1* background. 

### 2.3. Overexpressing PDAT1 Diverts Fatty Acid Flux from Membrane Lipids to TAG

PDAT1 catalyzes the transesterification of an acyl group from membrane phospholipids to diacylglycerol to produce TAG and a lysophospholipid [16]. To test whether an overexpression of PDAT1 affects membrane lipids, total lipid extracts from leaves of 4-week-old plants were separated by thin-layer chromatography (TLC) and individual lipids were quantified by gas chromatograph-mass spectrometry. Since the two independent lines consistently showed very similar growth behavior (Figure 1), TAG content (Figure 2) and fatty acid metabolic rates (Figure 3), only the transgenic line 19 was chosen for detailed membrane lipid profiling for clarity and simplicity. The results revealed significant decreases in levels of MGDG and DGDG in the overexpressing line, whereas other major membrane lipids remained largely unaltered, compared with those of *adg1* (Figure 4A). The fatty acid composition of major membrane lipids was also altered. Notably, in the PDAT1 overexpressing line compared with *adg1,* there were significant increases in 16:3 at the expense of 18:3 in both MGDG (Figure 4B) and DGDG (Figure 4C), increases in 16:0 and 18:1 DGDG (Figure 4C) and an increase in 18:1 in PC (Figure 4D). Since 16:3 is almost exclusively located at the *sn-2* position of galactolipids [22], the increase in 16:3 is indicative of an increased flux of fatty acids towards the plastid pathway of thylakoid lipid synthesis in PDAT1-overexpressing lines, consistent with our previous results [14]. 

### 2.4. Overexpressing PDAT1 Diverts Carbon Flux from Sugars to Lipids

In leaf tissues, fatty acid synthesis competes with sugars and starch for photosynthetically fixed carbon [23]. Therefore, the increased fatty acid synthesis in PDAT1 overexpressing lines in *adg1* may result in a decrease in sugar accumulation if photosynthesis is not increased accordingly. To test this possibility, several parameters associated with photosynthesis were compared. The maximum efficiency of PSII photochemistry (Fv/Fm) was similar between *adg1* and *adg1* lines overexpressing PDAT1 (Figure 5A). Likewise, there were no significant differences in the linear photosynthetic electron transport rate (ETR) (Figure 5B), photochemical quenching (qP) and quantum yield of photochemical energy conversion (Φ_PSII_) (Figure 5A). On the other hand, levels of total leaf sugars (Figure 6A), mainly glucose, sucrose and fructose (Figure 6B and 6C) measured at the end of the light period were significantly decreased in PDAT1 overexpressing lines compared with *adg1*. In both *adg1* and the transgenic lines, sugars accumulated during the day were almost completely depleted 4 h into the night period. Together, these results suggest that an overexpression of PDAT1 increases carbon flux into fatty acids at the expense of sugars.

## 3. Discussion

We recently showed that deficiency in starch synthesis in *adg1* results in increased rates of fatty acid synthesis and turnover without impacting overall membrane content [21]. Results from the present study show that overexpression of PDAT1, a critical enzyme catalyzing the last step in TAG synthesis in leaves, enhances fatty acid and TAG synthesis at the expense of soluble sugars and improves the growth of the starchless mutant *adg1*. Lipids contain more than twice as much energy per gram as carbohydrates and proteins. In addition, unlike sugars, which can be metabolized easily and quickly as a source of immediate energy, TAG breakdown is complex and slow, requiring the coordinated actions of several different subcellular organelles including lipid droplets, peroxisomes and mitochondria. The difference in the rate at which sugars and TAG are metabolized is clearly illustrated in *adg1* lines overexpressing PDAT1. Although, leaves of the PDAT1-overexpressing lines accumulated up to two-fold more sugars than TAG on a dry weight (DW) basis at the end of the day, sugars were completely depleted within the first 4 h into the night period (Figure 6), whereas as much as 30% TAG remained at the end of the dark period (Figure 1). Sugars were almost completely consumed within the initial 4 h of the night period in *adg1* (Figure 6), leaving *adg1* plants starved for carbon and energy for the rest of the night period, which has been shown to cause severe reductions in plant growth during the following light period [24]. Overexpression of PDAT1 diverts about half of carbon destined for sugar synthesis into TAG. The slow breakdown of TAG accumulated during the day allows a constant supply of fatty acids for energy production throughout the night period, which alleviates nighttime energy deficiency in the absence of transient starch, likely explaining why overexpression of PDAT1 improves the growth of *adg1*.

Leaf photosynthetic capacity is known to be regulated by the rate of utilization of photoassimilates in the rest of the plants [25]. Indeed, increasing sink strength has been demonstrated to increase the rate of photosynthetic carbon assimilation in wide variety of species [26,27]. Fatty acid synthesis is a highly energy-demanding process, consuming chemical energy stored in seven molecules of ATP and reducing power in 14 molecules of NADPH for every molecule of 16:0 produced [28], in addition to carbon precursors in the form of triose phosphate, the primary end product of photosynthetic carbon fixation. Therefore, it has be hypothesized that increasing TAG accumulation in leaves may lead to an increase in photosynthesis [29]. Our study shows that the overexpression of PDAT1 had no significant effects on photosynthesis as assessed by chlorophyll fluorescence kinetic parameters, and that TAG accumulation in PDAT1 overexpressing lines was at the expense of sugars. These results imply that under our growth conditions, photosynthesis is not limited by sink activity, but possibly by available light. Future studies will test whether increasing TAG accumulation affects photosynthesis and plant fitness and performance under high light conditions or in natural environments. Since leaves analyzed in this study may contain chloroplasts at different levels of development, further studies are also needed to assess the difference in the pathway of carbon flow as well as in response to changing light intensities between developing and mature chloroplasts.

## 4. Materials and Methods

### 4.1. Plant Materials and Growth Conditions

The *adg1* mutant used in this study was obtained from the Arabidopsis Biological Research Center at Ohio State University [30]. This homozygous mutant carries a point mutation in the gene encoding the large subunit of ADG1 [6]. For growth in soil, seeds were germinated on 0.6% (*w*/*v*) agar-solidified one-half-strength Murashige and Skoog (MS) medium [31] supplemented with 1% (*w*/*v*) sucrose in an incubator with a photon flux density of 50 to 80 μmol m^−2^ s^−1^, a light period of 16 h (22 °C), and a dark period of 8 h (18 °C). Ten-d-old seedlings were transferred to soil and grown under a photosynthetic photon flux density of 80 to 150 μmol m^−2^ s^−1^ at 22/18 °C (day/night) with a 16-h-light/8-h-dark period. For growth analysis, shoot biomass was measured from 20–30 plants per genotype. In each experiment, three random pots were used as three biological replicates, and the growth analysis experiment was conducted at least three times using independent trials with similar designs and similar results.

### 4.2. Generation of the Construct and Plant Transformation

The plant overexpression vector harboring *PDAT1* was generated as described in our previous publication [14]. The construct was transformed into *adg1* following the floral dip protocol with *Agrobacterium tumefaciens* strain C58C1 [32]. Transgenic plants were selected in the presence of antibiotics for the vector on MS medium. T_2_ progeny from independent transgenic lines containing a single T-DNA insertional event (based on antibiotic resistance properties) was chosen for detailed phenotypic analyses. 

### 4.3. Lipid Analysis

Samples for lipid analysis were taken from plants approximately 4 h into the light period, unless stated otherwise. Polar and neutral lipids were separated on silica plates (Silica Gel 60, EMD Millipore Corporation, Darmstadt, Germany) by thin layer chromatography (TLC) using acetone-toluent-water (91:30:7, by volume) and/or hexane-diethyl ether-acetatic acid (70:30:1, by volume), respectively. Fatty acid methyl esters were prepared as described by Li-Beisson et al. [13]. Separation and identification of the fatty acid methyl esters were performed on an HP5975 gas chromatograph-mass spectrometer (Hewlett-Packard) fitted with a 30 m × 250 μm DB-23 capillary column (Agilent, Santa Clara, CA) with helium as the carrier gas as described [14]. Fatty acid methyl esters were quantified using heptadecanoic acid as an internal standard using flame ionization analysis as described [14].

### 4.4. Radiotracer Labeling

Labeling experiments with ^14^C-acetate were done according to Koo et al. 2005 [33]. Briefly, rapidly growing leaves of four-week-old plants were incubated in the light of 80 µmol m^−2^ s^−1^ at 22 °C in 10 mL of medium containing 20 mM MES pH5.5, one-tenth strength of MS salts and 0.01% Tween 20. The assay was started by the addition of 0.1 mCi of ^14^C-acetate (106 mCi/mmol; American Radiolabeled Chemicals, St. Louis, MO, USA). After incubating for 1 h, leaves were washed two times with water and immediately used for lipid extraction. For pulse-chase labeling experiments, leaves were labeled for 1 h with ^14^C-acetate. After washing with water, the leaves were then incubated further with an unlabeled solution for three days. Total lipids were extracted and separated as described [14] and radioactivity associated with total lipids or different lipid classes was determined by liquid scintillation counting. 

### 4.5. Measurement of Chlorophyll Fluorescence Parameters

Chlorophyll fluorescence measurements were performed with a portable pulse amplitude fluorometer (PAM-2000; Walz) in four-week-old plants grown on soil. The Fv/Fm was recorded after 15 min of dark adaptation. The fluorescence kinetic parameters for ETR and Φ_PSII_ were recorded during steady state photosynthesis, which was achieved after 15 min of illumination with an actinic light at a PFD identical to that the plants experienced during growth. Φ_PSII_ was calculated according to the following equations: Φ_PSII_ = (F*_m_*′ − F′)/F*_m_*′ and qP = (F*_m_*′ − F)/(F*_m_*′ − F*_o_*′) [34], where F*_o_′,* F*_m_*′ and F′ are the minimum, maximal and steady-state fluorescence yields from light-adapted leaves, respectively. Photosynthetic ETR was calculated as the product of Φ_PSII_ × actinic light intensity × 0.84 × 0.5 according to Genty et al. [35]. 

### 4.6. Quantification of Sugars

Soluble sugars were extracted twice with 80% ethanol at 80 °C and extracts were dried and dissolved in distilled water. Levels of soluble sugars (glucose, fructose and sucrose) in ethanol extracts were quantified by the sequential addition of glucose-6-P dehydrogenase, hexokinase, glucose-6-P isomerase and invertase and measuring the increase in absorbance at 340 nm as described by Stitt et al. [36]. 

## 5. Conclusions

The results presented here show that lipids in the form of TAG can partially replace the function of starch in maintaining energy homeostasis and plant growth in starchless mutants. Overexpression of PDAT1 enhances fatty acid synthesis and turnover and diverts fatty acid flux from membrane lipids into TAG. During the nighttime, TAG hydrolysis occurs in a linear manner, providing a constant supply of fatty acids for energy production throughout the long night, and thus contributing to metabolism and growth in starchless mutants. The increased TAG accumulation due to PDAT1 overexpression is at the expense of sugars. Consistent with this, our chlorophyll fluorescence kinetic data showed that the increased sink strength due to PDAT1 overexpression does not lead to an increase in photosynthetic carbon assimilation, suggesting that under our experimental conditions, photosynthesis is limited by sources such as light intensity but not by sink demands.

## Figures and Tables

**Figure 1 plants-08-00229-f001:**
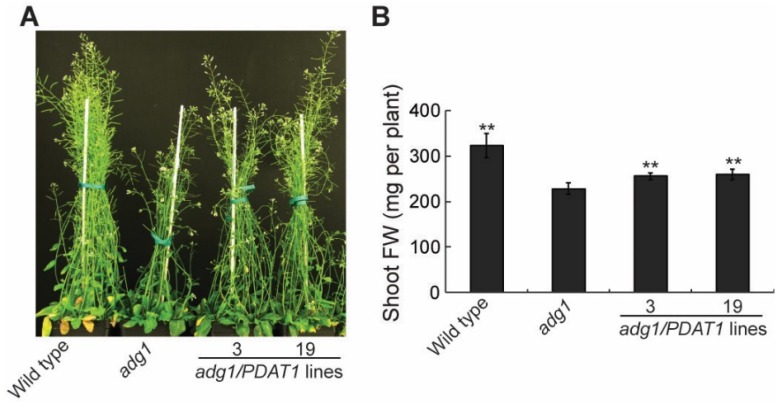
Overexpression of PDAT1 improves the growth of *adg1*. (**A**) Six-week-old plants grown on soil under long days. (**B**) Shoot biomass of six-week-old plants grown on soil. Asterisks indicate statistically significant differences from *adg1* based on Student’s *t* test (*P* < 0.01).

**Figure 2 plants-08-00229-f002:**
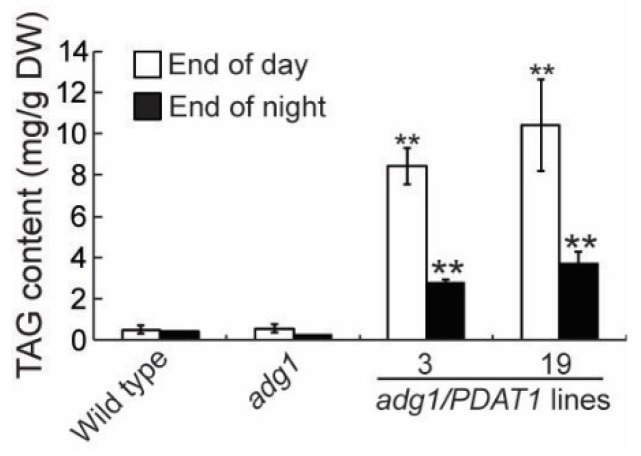
TAG content in leaves of six-week-old plants grown on soil. Asterisks indicate statistically significant differences from *adg1* based on Student’s *t* test (*P* < 0.01).

**Figure 3 plants-08-00229-f003:**
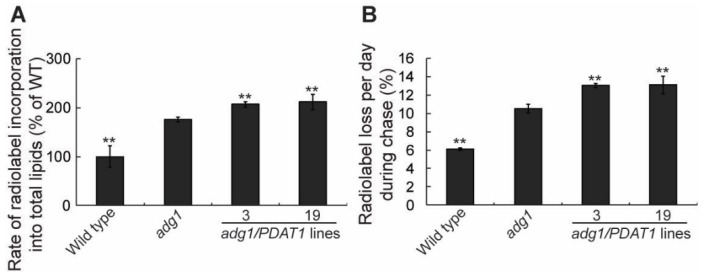
Overexpression of PDAT1 enhances both fatty acid synthesis and turnover in *adg1*. (**A**) Rate of ^14^C-acetate incorporation into total leaf lipids. (**B**) Radiolabel loss during 3 d of chase following 1 h of pulse with ^14^C-acetate. Asterisks indicate statistically significant differences from *adg1* based on Student’s *t* test (*P* < 0.01).

**Figure 4 plants-08-00229-f004:**
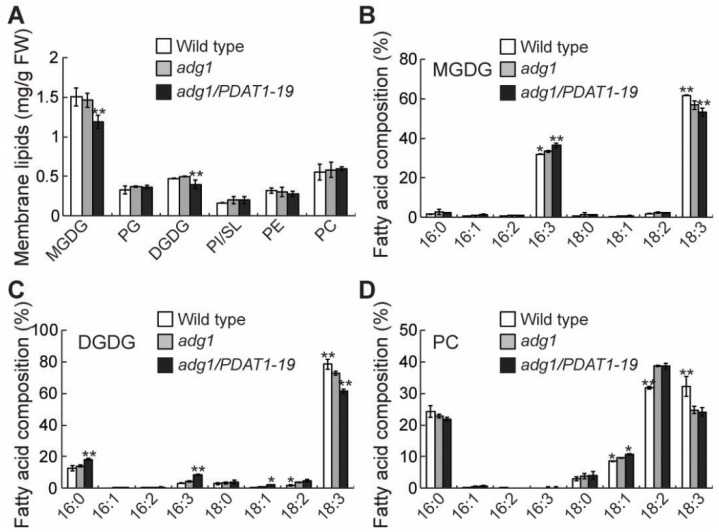
Membrane lipid content (**A**) and fatty acid composition of MGDG (**B**), DGDG (**C**) and PC (**D**) in leaves of six-week-old plants grown on soil. Asterisks indicate statistically significant differences from *adg1* based on Student’s *t* test (* *P* < 0.05, ** *P* < 0.01).

**Figure 5 plants-08-00229-f005:**
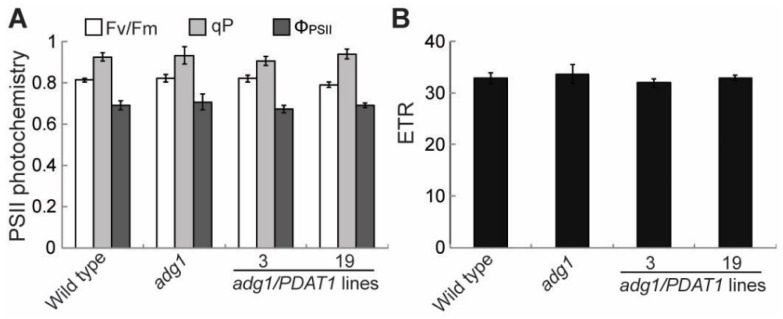
Chlorophyll fluorescence parameters in leaves of six-week-old plants grown on soil. (**A**) Photosystem II (PSII) photochemistry. (**B**) Photosynthetic electron transport rate (ETR).

**Figure 6 plants-08-00229-f006:**
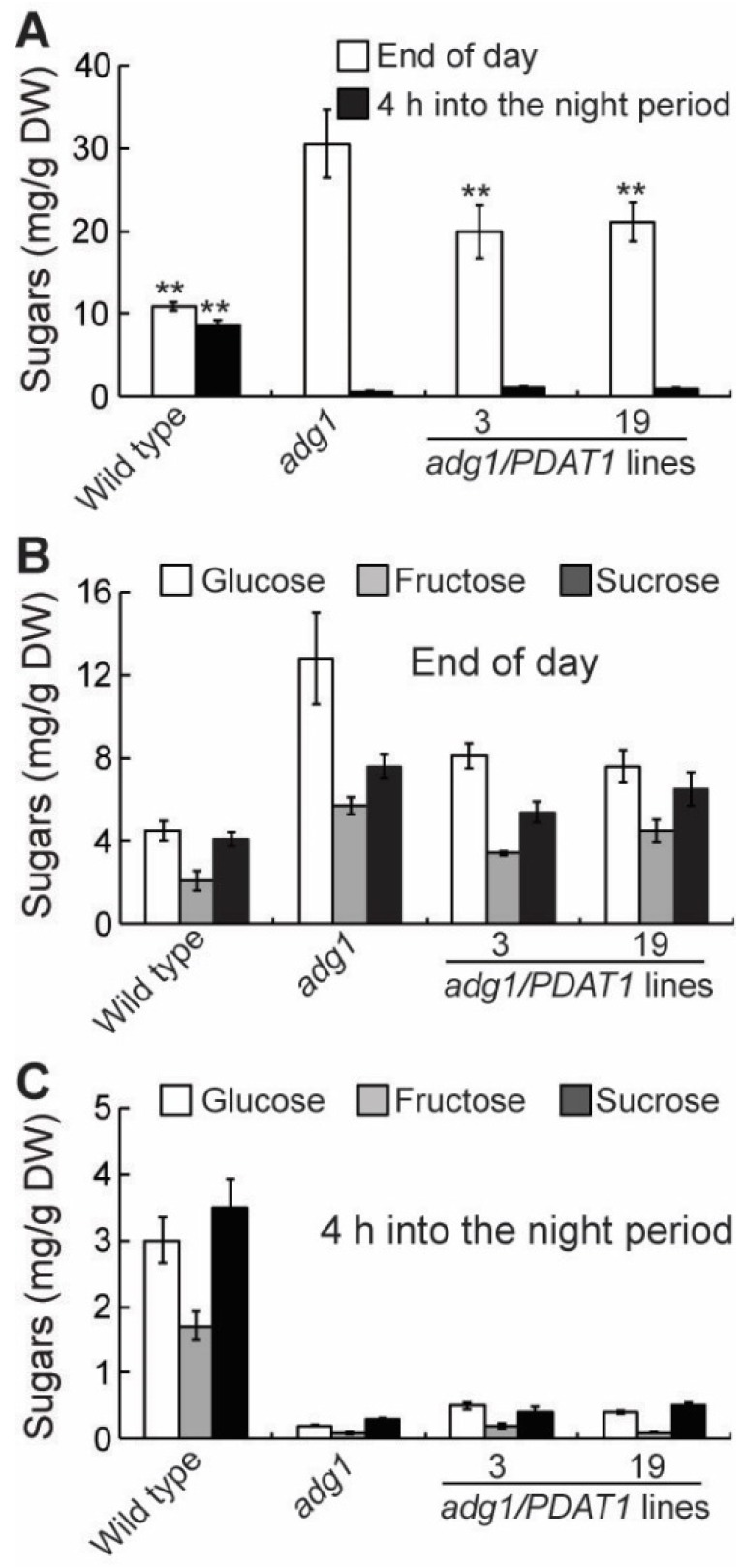
Sugar content in leaves of six-week-old plants grown on soil. (**A**) Total sugar content. (**B**) and (**C**) Glucose, fructose and sucrose levels. Asterisks indicate statistically significant differences from *adg1* based on Student’s *t* test (*P* < 0.01).

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
