# Peer review of "Diversion of Carbon Flux from Sugars to Lipids Improves the Growth of an Arabidopsis Starchless Mutant"

_plants, 2019, doi:10.3390/plants8070229_

Round 1

Reviewer 1 Report

The submitted work reports that the diversion of carbon flux from sugars to oils improved the growth of starchless mutant of Arabidopsis. The authors compared the growth between wild type, starchless mutant atg1, and two lines of PDAT1-overexpresing atg1 transgenic plants and found that triacylglycerol (TAG) accumulated by PDAT1 overexpression was able to compensate for the lack of starch during the growth under light/dark conditions. The authors also found that the PDAT1 overexpression also enhanced the fatty acid turnover and TAG breakdown was slower than sugar consumption during the night-time growth in atg1 and suggested that lipids can contribute to diurnal plant growth by supplying fatty acids as an energy source. This manuscript was well-written and straight forward without any problems. However, the reviewer has a major concern on the method of lipid quantification. As far as the authors showed as the references, heptadecanoic acid was used as an internal standard. If it is a case of GC-FID, which means that the detection system is not mass-spec but FID, single internal standard works as a standard for quantification. However, in case of mass-spec, the extent of ionization of the FAME depends on the molecular specie, and thus quantification of FAME using mass-spec requires measurement of every single FAME as a standard. The authors should refer the manuscript (Jouhet et al 2017 PLOS one, LC-MS/MS versus TLC plus GC methods: Consistency of glycerolipid and fatty acid profiles in microalgae and higher plant cells and effect of a nitrogen starvation) and consider to show the lipid content written in the manuscript (Fig. 2 and Fig. 4) as a mass-spec signal intensity of each FAME which allows us to compared the amounts between the same FAME.

Minor points:

Line 162, the capital letter of “Significantly” should be a small letter.

Author Response

he submitted work reports that the diversion of carbon flux from sugars to oils improved the growth of starchless mutant of Arabidopsis. The authors compared the growth between wild type, starchless mutant atg1, and two lines of PDAT1-overexpresing atg1 transgenic plants and found that triacylglycerol (TAG) accumulated by PDAT1 overexpression was able to compensate for the lack of starch during the growth under light/dark conditions. The authors also found that the PDAT1 overexpression also enhanced the fatty acid turnover and TAG breakdown was slower than sugar consumption during the night-time growth in atg1 and suggested that lipids can contribute to diurnal plant growth by supplying fatty acids as an energy source. This manuscript was well-written and straight forward without any problems. However, the reviewer has a major concern on the method of lipid quantification. As far as the authors showed as the references, heptadecanoic acid was used as an internal standard. If it is a case of GC-FID, which means that the detection system is not mass-spec but FID, single internal standard works as a standard for quantification. However, in case of mass-spec, the extent of ionization of the FAME depends on the molecular specie, and thus quantification of FAME using mass-spec requires measurement of every single FAME as a standard. The authors should refer the manuscript (Jouhet et al 2017 PLOS one, LC-MS/MS versus TLC plus GC methods: Consistency of glycerolipid and fatty acid profiles in microalgae and higher plant cells and effect of a nitrogen starvation) and consider to show the lipid content written in the manuscript (Fig. 2 and Fig. 4) as a mass-spec signal intensity of each FAME which allows us to compared the amounts between the same FAME.

Response: Thank the reviewer for the valuable information and comments.

With regards to FAME quantification, our GC/MS is equipped with two detectors: an FID and a Triple_Axis Detector. All FAME data present in this work were quantified using the FID detector. This is now mentioned in the Method section in the revised manuscript.   

Minor points:

Line 162, the capital letter of “Significantly” should be a small letter.

Response: Corrected. Thank you.

Reviewer 2 Report

Congratulations, this is an excellent manuscript.

I have attached two comments. But they mostly apply to a following publication rather than this manuscript.

Keeping in mind earlier findings (especially those published by Foyer and Stitt), it may be discussed, at what extend your findings depend on the experimetal conditions used in your experiments.

Author Response

Thank the reviewer for constructive comments. We have discussed these important points in the Discussion section of the revised manuscript, which reads:

"Since leaves analyzed in this study may contain chloroplasts at different levels of development, further studies are also needed to assess the difference in the pathway of carbon flow as well as in response to changing light intensities between developing and mature chloroplasts." 

Reviewer 3 Report

The manuscript entitled “Diversion of Carbon Flux from Sugars to Lipids Improves the Growth of an Arabidopsis Starchless Mutant describes the effect of the overexpression of phospholipid:diacylglycerol  acyltransferase1 (PDAT1)  in Arabidopsis thaliana adg1 mutant with starchless phenotype. The authors demonstrate that PDAT1 overexpression enhances triacylglycerol (TAG) accumulation at the expense of sugars. While sugars are consumed in four hours during night, the dynamics of TAG degradation is slower. Thus, PDAT1 overexpression in adg1 background enables a better growth alleviating the energy  shortness at night. The topic of this work is interesting, the manuscript is well written, the results are clearly presented. In my opinion, the manuscript can be published after some essential revisions.

Abstract

Lines 16 and 23  I’d suggest growth instead of diurnal growth

Introduction

Line 40      “…. but similar growth rates to wild type under continuous night” is it correct?

Results

Fig.1B  The increase in fresh weight in PDAT1 overexpressing lines versus adg1 is statistically significant, but slight. The authors should indicate in the results the percentage of increase.

Fig.4 Only one PDATI1 overexpressing line (19) was analysed.  Why?

Fig.6 I would expect a more detailed analysis of sugars distinguishing between sucrose and exoses

 Material and methods

Line 201  The mutant  line although already cited, should be described in more details (type of mutation, zygosity).

Also, the two overexpressing lines should be better described. Are single copy insertions? Are the lines homozygous?

The behaviour of the adg1 mutant should be discussed, the mutant shows a higher synthesis and turnover of fatty acids in comparison with the wt, similarly to the PDAT1 overexpressing lines, and  membrane lipid content unchanged.

Author Response

Thank the reviewer for valuable comments. Our point-by-point responses are shown below: 

Lines 16 and 23  I’d suggest growth instead of diurnal growth

Response: "Diurnal" has been deleted in the abstract.

Introduction

Line 40      “…. but similar growth rates to wild type under continuous night” is it correct?

Response: Yes.

Results

Fig.1B  The increase in fresh weight in PDAT1 overexpressing lines versus adg1 is statistically significant, but slight. The authors should indicate in the results the percentage of increase.

Response: The percent of increase is now described in the Result section.

Fig.4 Only one PDATI1 overexpressing line (19) was analysed.  Why?

Response: The reason as to why only one line was used in the analysis is now described in the Result section, which reads:

"Since the two independent lines consistently showed very similar growth behavior (Figure 1), TAG content (Figure 2) and fatty acid metabolic rates (Figure 3), only the transgenic line 19 was chosen for detailed membrane lipid profiling for clarity and simplicity.

Fig.6 I would expect a more detailed analysis of sugars distinguishing between sucrose and exoses

Response: The data are now shown in Figures 6B and 6C.

 Material and methods

Line 201  The mutant  line although already cited, should be described in more details (type of mutation, zygosity).

Response: We have provided a more detailed description of the mutant in the Method section, which reads:

"This homozygous mutant carries a point mutation in the gene encoding the large subunit of ADG1 [6]." 

Also, the two overexpressing lines should be better described. Are single copy insertions? Are the lines homozygous?

Response: We have provided a more detailed description of the overexpressing lines in the method section, which reads:

"T2 progeny from independent transgenic lines containing a single T-DNA insertional event (based on antibiotic resistance properties) was chosen for detailed phenotypic analyses. "

The behaviour of the adg1 mutant should be discussed, the mutant shows a higher synthesis and turnover of fatty acids in comparison with the wt, similarly to the PDAT1 overexpressing lines, and  membrane lipid content unchanged.

Response: We have discussed the adg1 mutant in the Discussion section, which reads:

" We recently showed that deficiency in starch synthesis in adg1 results in increased rates of fatty acid synthesis and turnover without impacting overall membrane content [21]."